# A Fast-Response Red Shifted Fluorescent Probe for Detection of H_2_S in Living Cells

**DOI:** 10.3390/molecules25030437

**Published:** 2020-01-21

**Authors:** Ismail Ismail, Zhuoyue Chen, Xiuru Ji, Lu Sun, Long Yi, Zhen Xi

**Affiliations:** 1State Key Laboratory of Elemento-Organic Chemistry and Department of Chemical Biology, National Engineering Research Center of Pesticide (Tianjin), College of Chemistry, Nankai University, Tianjin 300071, China; ismailics86@gmail.com; 2Beijing Key Laboratory of Bioprocess and College of Chemical Engineering, Beijing University of Chemical Technology, 15 Beisanhuan East Road, Chaoyang District, Beijing 100029, China; 2018210060@mail.buct.edu.cn; 3Tianjin Key Laboratory on Technologies Enabling Development of Clinical Therapeutics and Diagnostics (Theranostics), School of Pharmacy, Tianjin Medical University, Tianjin 300070, China; jxry1217@tmu.edu.cn (X.J.); sunlu@tmu.edu.cn (L.S.); 4Collaborative Innovation Center of Chemical Science and Engineering (Tianjin), Tianjin 300071, China

**Keywords:** fluorescent probe, red shifted, H_2_S, bioimaging

## Abstract

Near-infrared (NIR) fluorescent probes are attractive tools for bioimaging applications because of their low auto-fluorescence interference, minimal damage to living samples, and deep tissue penetration. H_2_S is a gaseous signaling molecule that is involved in redox homeostasis and numerous biological processes in vivo. To this end, we have developed a new red shifted fluorescent probe **1** to detect physiological H_2_S in live cells. The probe **1** is based on a rhodamine derivative as the red shifted fluorophore and the thiolysis of 7-nitro 1,2,3-benzoxadiazole (NBD) amine as the H_2_S receptor. The probe **1** displays fast fluorescent enhancement at 660 nm (about 10-fold turn-ons, *k*_2_ = 29.8 M^−1^s^−1^) after reacting with H_2_S in buffer (pH 7.4), and the fluorescence quantum yield of the activated red shifted product can reach 0.29. The probe **1** also exhibits high selectivity and sensitivity towards H_2_S. Moreover, **1** is cell-membrane-permeable and mitochondria-targeting, and can be used for imaging of endogenous H_2_S in living cells. We believe that this red shifted fluorescent probe can be a useful tool for studies of H_2_S biology.

## 1. Introduction

Recently biological reports have demonstrated the important role of H_2_S, and the results suggested that endogenously-produced hydrogen sulfide (H_2_S) has been marked as gasotransmitter, allowing the regulation of numerous important physiological functions including; cardiovascular, gastrointestinal, endocrine, nervous, and immune systems [1,2,3]. Generally, endogenous H_2_S can be produced enzymatically from L-cysteine (Cys) by means of three distinctive enzymatic pathways; cystathionine γ-lyase (CSE), cystathionine β-synthetase (CBS), and 3-mercaptopyruvate sulfurtransferase (3-MST) [4]. The interest in the molecular mechanisms of H_2_S associated with physiology and pathology was sparked out of its recognition as a vital signaling molecule. However, the abnormal levels of H_2_S production lead to number of different human diseases including; diabetes [5], Alzheimer’s disease [6] liver cirrhosis [7], and the symptoms of Down’s syndrome [8,9,10]. As an important role played by H_2_S in tumor biology, it is proposed that the both production and inhibition of H_2_S concentration beyond a threshold level could exert anticancer effects [10,11]. While in plants, the growth and development, seed germination, and stress tolerance including cross-adaptation are regulated by H_2_S [12,13,14]. H_2_S also plays important role in microorganisms [15]. Due to its wide distribution in all organisms, the physiological characters of H_2_S and the precise mechanisms by which H_2_S may involve in vivo still remain largely unexplored. Therefore, adequate tools (H_2_S probes or donors) are necessary to further explore H_2_S biology [16,17,18,19,20,21]. For cellular H_2_S detection, various fluorescence probes have been successfully developed [22,23,24,25,26,27,28,29,30,31,32,33,34,35,36,37,38]. However, H_2_S fluorescence probes for in vivo bioimaging are still rare [16,17,18], especially for the imaging of H_2_S-related diseases including cancers [22]. Due to the potential limitations of fluorescent probes in deep tissue penetration, photodamage to biological samples, and background auto-fluorescence in living systems, it is crucial to develop probes associated with long wavelength emission especially in the red shifted region [39,40,41,42].

Recently, numbers of red shifted and NIR fluorescent dyes have been developed [43,44,45,46,47,48,49,50,51]. Consequently, various NIR fluorescent probes have been designed on the basis of different organic reaction scenario [8,52,53,54,55,56,57,58]. Among them, we as well as others discovered a reaction of H_2_S-specifc thiolysis of 7-nitro 1,2,3-benzoxadiazole (NBD) amines [33,58,59] to detect millimolar H_2_S in a long range wavelength. Herein, this reaction strategy is further employed for the development of a new red shifted fluorescent probe 1 (Scheme 1).

NIR dyes, including cyanine (Cy), are considered the classic NIR fluorescent dyes [60]. However, due to their flexible molecular structure, some of NIR fluorescent dyes accompanied some short comings for example, small Stokes shift, limited fluorescence quantum yield, low photo-stability lying, and high occupied molecular orbital (HOMO) energy levels [61,62]. Such photo-physical properties strongly affect the fluorescence signals due to the high background signal, which in turn result in low contrast for bioimaging [63,64,65,66]. In 2017, we developed a Cy-NBD probe (Scheme 1) which has the limitation of low quantum yield after H_2_S activation [58]. On the other side, classical rhodamine dyes contributed much in the field of biomolecular detection and biomedical imaging because of its magnificent photophysical and chemical properties [30,67,68,69,70,71]. Due to limited π-conjugated system of xanthene core derivatives, such as rhodamine B, rhodamine 6G, and rhodamine 123 have their emission wavelengths in the visible region (<600 nm). Recently, significant advancements have been made in the improvement of rhodamines-based fluorescent dyes with extended the π-conjugated system possessing long emission wavelength, high fluorescence quantum yield, and outstanding photostability [70,71,72,73,74,75]. Herein, we report the development of an extended π-conjugated rhodamine-NBD based probe **1** for the highly selective imaging of endogenous H_2_S in a red shifted region [70,71]. The probe is high selectivity towards H_2_S among other biothiols, with fluorescence emission in the red shifted region (>660 nm) and high fluorescence quantum yield (0.29) after H_2_S activation. The probe is successfully used for bioimaging of endogenous H_2_S in living cells.

## 2. Results and Discussion

### 2.1. Synthesis of ***1***

Probe **1** was constructed using a three-step route with a good yield (Scheme 2). By using the procedure described in the literature [76], 6-(dimethylamino)-3,4-dihydronaphthalen-1 (2H)-one **2** was first synthesized from commercially available 6-amino-3,4-dihydronaphthalen-1(2H)-one, which was then transformed to **3**. Finally, the probe **1** in 79% yield, was prepared by the coupling of compound **3** with NBD-piperazine. The facile and economic synthesis is important for the wide use of the probe. The structure of compound **1** was confirmed by ^1^H NMR, ^13^C NMR, and high resolution mass spectrum (HRMS). The spectra (Appendix A) are included in the Appendix A.

### 2.2. UV-Vis and Fluorescence Response of **1** towards H_2_S

With the probe in hand, we first tested the solubility of **1** in buffer solution. The linearity of **1** verified its good solubility up to 20 µM (Appendix A). Further, we tested the optical properties of **1**, and the absorbance and emission profiles are illustrated in Figure 1. As shown in Figure 1A, **1** displayed UV absorbance maxima at 620 nm and 500 nm, which are assigned to the rhodamine and NBD absorbance respectively. Since Na_2_S is a well-known inorganic H_2_S donor that is widely employed in the study of H_2_S effects on physiology, we used it as a H_2_S equivalent [77]. When reacted with H_2_S, the increase in intensity of absorbance peaks appeared between 600 nm and 520 nm, which could be assigned to the yielding of **4** and NBD-SH. The reaction between **1** and 500 µM H_2_S in PBS buffer (50 mM, pH 7.4) finished within 5 min. Furthermore, such thiolysis reaction was characterized by NMR with the formation of NBD-SH peaks (Appendix A) and HRMS with the production of peak at 535.3070 (calculated value for [4]^+^: 535.3068) (Appendix A).

The probe **1** showed weak fluorescence (quantum yield ɸ, 0.021) upon excitation at 620 nm, indicating that fluorescence in **1** could be mainly quenched by the photoinduced electron transfer process (PET) effect from the NBD moiety [58]. When **1** reacted with H_2_S, an excellent fluorescence change in a red shifted range with high brightness was observed (Figure 1B) with 10-fold turn-ons at 660 nm, and the quantum yield of the red shifted product was 0.29. The absorbance and emission data suggested the stokes shift up to 40 nm in PBS. A large Stokes shift could reduce the risk of background fluorescence and thus avoid self-quenching and backscattering effect upon excitation. These preliminary studies suggest the extended π-conjugated system of rhodamines provides excellent red shifted fluorescence probes for detection in long range with high brightness.

### 2.3. Kinetics Studies

Reaction kinetics, as an important parameter, was investigated for the probe **1** with H_2_S on account of its biological applicability under physiological conditions. To this end, the time-dependent fluorescence at 660 nm was recorded for data analysis (Figure 2A). The pseudo-first-order rate, k_obs_, was found by fitting the data with a single exponential function. Plotting log[H_2_S] versus log[k_obs_] confirmed a first-order dependence in H_2_S (Figure 2B). The reaction rate k_2_ (29.8 M^−1^s^−1^) was obtained by linear fitting of the k_obs_ versus H_2_S concentration (Figure 2C). The H_2_S-reaction rate of **1** is faster than our previous Rh-NBD-based probe [30], implying that such NBD-based probes can be employed for fast detection of H_2_S. On the other hand, HPLC was further employed to identify the fast reaction of **1** with H_2_S (Appendix A). Furthermore, fluorescent titration (Figure 3) was performed to determine the limit of detection (LOD) of **1** for H_2_S as 0.27 μM by using the 3σ/k method [58].

### 2.4. Selectivity and Co-Interference Studies

With above promising outcomes, we further investigated the selectivity and sensitivity of probe **1**. The fluorescent ‘‘off–on’’ response of **1** towards biothiols was measured. Probe **1** (2 μM) was treated with Cys, Hcy, and GSH individually (each 1 mM). As shown in Figure 4, the results showed that fluorescence intensity enhancement for analytes was nearly negligible except H_2_S, suggesting that **1** can selectively sense H_2_S. In order to check the interference of biothiols with coexistent H_2_S, we also tested **1** with these analytes in the presence of H_2_S (Figure 4). These findings suggested that all analytes did not interfere the H_2_S-specific thiolysis reaction. Furthermore, pH-dependent experiments were carried out to check whether **1** could sense at physiological pH (Appendix A). Obviously, the fluorescence enhancement occurred at pH 7.0–9.0, implying that **1** could work efficiently at physiological conditions.

### 2.5. Imaging of Probe ***1*** in Living Cells

Encouraged by the above results, we moved forward to study the biological applications of **1**. The cytotoxicity of the **1** was evaluated firstly by using the normal human umbilical vein endothelial cell (HUVEC) line via a standard MTT assay (Appendix A). After 24 h incubation with a varied concentration range of **1** from 5 µM, over 85% of the cells still remained viable, implying the relatively good biocompatibility of **1**.

To examine the application potential of **1** for H_2_S detection in living cells, HeLa cells were chosen as the model biological system. Briefly, the cells were incubated with **1** alone or co-incubated with **1** and Na_2_S/D-Cys for 30 min. Then, all cells were examined via the confocal microscopy. Cells with probe **1** treatment displayed faint fluorescence (Figure 5E), while cells displayed remarkable red fluorescence in the presence of **1** and Na_2_S (Figure 5F). These results demonstrated that **1** could be used for selective imaging of exogenous H_2_S. For detection of endogenous production of H_2_S, cells were co-incubated with D-Cys and **1**, as D-Cys can induce H_2_S biosynthesis via the 3-MST pathway [4]. Strong fluorescence was observed in cells (Figure 5G), which revealed that the endogenous production of H_2_S from D-Cys could be detected by **1**. To further confirm the detection of endogenous production of H_2_S from D-Cys by **1**, an inhibitor (aminooxyacetic acid, AOAA) was introduced to block the pathway for H_2_S production from D-Cys [4]. No obvious fluorescence was detected in the AOAA-treated cells (Figure 5H). These preliminary studies suggested that probe **1** could be used for visualization of H_2_S in cells efficiently and selectively.

The probe **1** contains a positive charge, which might be mitochondria-targeting [33]. To this end, a fluorescent co-localization assay with Mito-Tracker Green FM (a well-known mitochondria specific dye) and probe **1** with D-Cys was carried out. As shown in Figure 6, the green fluorescence signal produced by Mito-Tracker Green FM and the red fluorescence signal from probe **1** merged well in the cells (Figure 6C). The Pearson’s coefficient is 0.946. These data implied that the probe **1** is a promising tool for imaging of mitochondria H_2_S.

## 3. Experimental

### 3.1. Materials and Methods

All chemicals and solvents used for the synthesis were purchased from commercial suppliers and applied directly in the experiments without further purification. The progress of the reaction was monitored by TLC on pre-coated silica plates (60F-254, 250 μm) in thickness (Merck, Darmstadt, Germany), and spots were visualized by basic KMnO_4_, UV light or iodine. Merck silica gel 60 (100–200 mesh) was used for general column chromatography purification. ^1^H NMR and ^13^C NMR spectra were recorded on a Bruker 400 spectrometer (Karlsruhe, Germany). Chemical shifts are reported in parts per million with respect to the internal standard tetramethylsilane (Si(CH_3_)_4_ = 0.00 ppm) or residual solvent peaks (CD_2_Cl_2_ = 5.32 ppm; CDCl_3_ = 7.26 ppm; DMSO-d_6_ = 2.5 ppm). ^1^H NMR coupling constants (J) are reported in hertz (Hz), and multiplicity is indicated as the following: s (singlet), d (doublet), t (triplet), dd (doublet of doublets), m (multiple). High-resolution mass spectra (HRMS) were obtained on an Agilent 6540 UHD Accurate-Mass Q-TOF LC/MS or Varian 7.0 T FTICR-MS. The UV-visible spectra were recorded on a UV-3600 UV-VIS-NIR spectrophotometer (Shimadzu, Japan). The fluorescence study was carried out using an F-280 spectrophotometer (Tianjin Gangdong Sci & Tech., Development. Co., Ltd. Tianjin, China).

### 3.2. Synthesis of 6-(dimethylamino)-3,4-Dihydronaphthalen-1(2H)-One

To a mixture of 6-aminotetralone (0.50 g, 3.1 mmol), CH_3_I (0.06 g, 4.6 mmol) and K_2_CO_3_ (1.3 g, 9.3 mmol) in DMF (5 mL) was stirred for 24 h at 40–45 °C. After completion of reaction, the mixture was cooled to room temperature, water (10 mL) was added and the solution was extracted with EtOAc (3 × 50 mL). The organic layers were combined, dried with anhydrous MgSO_4_ and the solvent was removed under reduced pressure. The residue was purified by silica gel column chromatography using PE (petroleum ether): EtOAc = 6:1 as the eluent to obtain pure compound (Yield: 53.0%, 0.31 g). ^1^H NMR (400 MHz, CDCl_3_) δ 7.95 (dd, J = 8.8, 1.6 Hz, 1H), 6.62 (d, J = 8.8 Hz, 1H), 6.43 (s, 1H), 3.06 (s, 6H), 2.88 (t, J = 5.2 Hz, 2H), 2.59–2.55 (m, 2H), 2.10–2.05 (m, 2H). ^13^C NMR (101 MHz, CDCl_3_) δ 196.9, 153.5, 146.6, 129.4, 121.9, 110.3, 109.6, 40.2, 38.9, 30.7, 23.7. HRMS [C_12_H_16_NO]^+^: Calcd. for [M+H]^+^ 190.1232; found: [M+H]^+^ 190.1228.

### 3.3. Synthesis of Intermediate ***3***

2-(4-diethylamino-2-hydroxybenzoyl)-benzoic acid (0.151 g, 0.48 mmol) and 6-(dimethylamino)-3,4-dihydronaphthalen-1(2H)-one (0.091 g, 0.48 mmol) were added to concentrated H_2_SO_4_ (10 mL) at 0 °C. The mixture was stirred at 100 °C for 2 h. After completion of reaction the mixture was allowed to cool at room temperature and was poured onto ice (10 g). HClO_4_ (1 mL) was gently added to the solution and the resulting precipitate was filtered off and washed with cold water. After drying, the residue was purified by silica gel column chromatography using CH_2_Cl_2_: methanol = 20:1 as the eluent to obtain pure compound as a purple-green solid (Yield: 80.0%, 0.17 g). ^1^H NMR (400 MHz, DMSO-d_6_) δ 13.22 (s, 1H), 8.19 (dd, J = 7.2, 2.4 Hz, 2H), 7.86 (t, J = 7.2 Hz, 1H), 7.75 (t, J = 7.6 Hz, 1H), 7.41 (d, J = 7.2 Hz, 1H), 7.24 (d, J = 1.6 Hz, 1H), 7.09 (d, J = 9.6 Hz, 1H), 6.93 (dd, J = 9.2, 2.0 Hz, 1H), 6.87 (d, J = 9.2 Hz, 1H), 6.75 (s, 1H), 3.58 (q, J = 6.4 Hz, 4H), 3.18 (s, 6H), 2.93–2.80 (m, 2H), 2.49–2.35 (m, 2H), 1.19 (t, J = 6.9 Hz, 6H). ^13^C NMR (101 MHz, DMSO-d_6_) δ 166.5, 163.4, 156.1, 155.0, 153.1, 145.1, 134.4, 133.0, 130.9, 130.0, 129.2, 129.1, 128.6, 118.4, 115.2, 114.3, 113.0, 112.0, 110.7, 96.0, 44.8, 40.0, 26.9, 23.6, 12.4. HRMS [C_30_H_31_N_2_O_3_]^+^: Calcd. for [M]^+^ 467.2329; found: [M]^+^ 467.2331.

### 3.4. Synthesis of Probe ***1***

Dissolved compound **2** (0.121 g, 0.2 mmol) in 5 mL DMF, followed by the addition of HATU (0.122 g, 0.32 mmol) and DIPEA (102 μL, 0.75 mmol). Stirred the solution for 5 min, NBD-piperazine (0.064 g, 0.2 mmol) was added to the solution and continue the stirring for 12 h at room temperature. After completion of reaction DMF was removed in vacuo. The residue was purified by silica gel column chromatography to give dark-red solid **1** (0.12 g, 79%). ^1^H NMR (400 MHz, DMSO-d_6_) δ 8.50 (d, J = 8.8 Hz, 1H), 8.16 (d, J = 8.8 Hz, 1H), 7.74 (bs, 3H), 7.48 (bs, 1H), 7.20 (bs, 1H), 7.07 (bs, 2H), 6.91 (d, J = 8.0 Hz, 1H), 6.72 (s, 1H), 6.54 (d, J = 9.2 Hz, 1H), 4.22–4.14 (m, 2H), 4.03–3.83 (m, 2H), 3.82–3.46 (m, 8H), 3.19 (s, 6H), 2.98–2.78 (m, 2H), 2.62–2.45 (m, 2H), 1.16 (s, 6H). ^13^C NMR (101 MHz, DMSO-d_6_) δ 167.0, 163.8, 156.0, 155.2, 153.0, 145.5, 145.3, 144.7, 136.2, 134.2, 131.9, 130.2, 129.5, 129.3, 129.3, 129.2, 127.6, 121.5, 119.6, 114.9, 114.3, 113.0, 112.1, 110.7, 103.4, 96.1, 49.0, 48.1, 45.6, 44.8, 40.8, 27.0, 23.9, 12.4. HRMS [C_40_H_40_N_7_O_5_]^+^: Calcd. for [M]^+^ 698.3085; found: [M]^+^ 698.3090.

### 3.5. Procedure for Spectroscopic Studies

All spectroscopic measurements were performed in phosphate-buffered saline buffer (PBS, 50 mM, pH 7.4, containing 10% DMSO) at room temperature. Compounds were dissolved into DMSO to prepare the stock solutions with a concentration of 5 mM. 1–500 mM Stock solutions of Na_2_S in degassed (by bubbling N_2_ for 30 min) PBS buffer were used as H_2_S source. Probes were diluted in PBS buffer (50 mM, pH 7.4, containing 10% DMSO) to afford the final concentration of 2–5 µM. For the selectivity experiment, different biologically relevant molecules (100 mM) were prepared as stock solutions in degassed PBS buffer. Appropriate amount of biologically relevant species was added to separate portions of the probe solution and mixed thoroughly. All measurements were performed in a 3 mL corvette with 2 mL solution. The reaction mixture was shaken uniformly before emission spectra were measured.

### 3.6. Cell Culture and Cytotoxicity Assay

The HUVEC and HeLa cell lines were purchased from the Cell Bank of the Chinese Academy of Sciences (Shanghai, China). And the cells were cultured in RPMI 1640 medium with 10% fetal bovine serum and 1% penicillin/streptomycin under standard cell culture conditions at 37 °C in a humidified CO_2_ incubator. Before the cytotoxicity assay, the HUVEC cells were transferred to the 96-well plate and cultured for one night. After that, the culture medium was replaced with a fresh one and the HUVEC cells were pre-incubated with probe **1** with a concentration range of 5–25 μM for 24 h. The cell viability was then measured by the standard MTT assay.

### 3.7. Cell Imaging

Glass bottom dishes were added into a 24-well plate for cell imaging before cells were seeded. Then, the HeLa cells were transferred to the 24-well plate and cultured for one night before the experiments. After that, the culture medium was replaced with the fresh one and the cells were treated with the desired reagents. After incubation, the HeLa cells were quickly washed with PBS three times, and then fixed with 4% paraformaldehyde solution for 10 min. Finally, the HeLa cells were washed with PBS and imaged using a confocal microscope (Olympus FV1000) with a 40× objective lens. Emission was collected at the green channel (500–530 nm, excitation at 488 nm) and the red channel (620–660 nm, excitation at 594 nm).

## 4. Conclusions

In summary, we have developed a new, extended π-conjugation rhodamine-NBD a red shifted fluorescence probe **1** capable of detection H_2_S in live cells. The probe shows a relatively large Stokes shift (40 nm), fast response (*k* = 29.8 M^−1^s^−1^), and good quantum yield (ø = 0.29) after H_2_S activation. Moreover, **1** was water-soluble, cell-membrane-permeable, and had high selectivity and sensitivity for H_2_S. We believe that this red shifted range probe **1** could be a useful tool for studies of H_2_S biology in the future.

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
