# Peer review of "A Fast-Response Red Shifted Fluorescent Probe for Detection of H_2_S in Living Cells"

_molecules, 2020, doi:10.3390/molecules25030437_

Round 1

Reviewer 1 Report

These article described probe which worked according well-known mechanism of thiolysis of 7-nitro-1,2,3-benzoxadiazole (NBD) amines. In this context, connection of fluorophore to the NBD is not a an innovative approach. In my opinion, this article should have some new aspects of application of probe in cells.

It is known that the analytes detection in cell culture that is based only on fluorescence imaging could lead to misinterpretation. Therefore, the best method which allow to prove that for example H2S is responsible for fluorescence, is to detect specific product formed in the reaction of probe with H2S.

In my opinion in this studied probe authors should detect NBD-SH as such fingerprint in buffer intracellularly and in the cell media. Moreover, it will be good to show the influence of the H2S quencher for spectroscopic response of probe and formation of NBD-SH.

It will be good to show the probe response during the enzymatic release of H2S in real time.

The article also lacks a comparison of properties of the described probe with other probes operating according to the same mechanism. (fluorescence enhancements, reaction rate, limit of the detection). 

Before acceptance of this paper some technical aspects should be clarified;

Page 7; line 202 and page 8 line 216. How many milliliters of water were added? Page 7; line 204. What does PE abbreviation mean? Page 8; line 214. How much ice was added? Page 8; line 249. How was the Na2S concentration verified? Page 8; line 245. What was the incubation time of analyte in solution with probe before the spectra were recorded? Page 9; line 266. Maybe it's better to write "cell cultures"

Author Response

This article described probe which worked according well-known mechanism of thiolysis of 7-nitro-1,2,3-benzoxadiazole (NBD) amines. In this context, connection of fluorophore to the NBD is not an innovative approach. In my opinion, this article should have some new aspects of application of probe in cells.

Answer: Thank you very much for this comment. Normally, rhodamine B, rhodamine 6G, and rhodamine 123 have their emission wavelengths in the visible region (< 600 nm). Here we report the development of an extended π-conjugated rhodamine-NBD based probe 1, and subsequently applied to detect and image biological H2S (Figure 5) in NIR region (> 650 nm) with high fluorescence quantum yield (0.29) after H2S activation.

1) It is known that the analytes detection in cell culture that is based only on fluorescence imaging could lead to misinterpretation. Therefore, the best method which allow to prove that for example H2S is responsible for fluorescence, is to detect specific product formed in the reaction of probe with H2S.

Answer: Thank you very much for this comment. In our previous works, we have indicated that the NBD amine-based probes are highly selective to H2S, which could be used for in vivo imaging of endogenous H2S (Ref. 31, 33, 58). Therefore, the detection of fluorescence could be used for bioimaging of H2S in this work.

2) In my opinion in this studied probe authors should detect NBD-SH as such fingerprint in buffer intracellularly and in the cell media. Moreover, it will be good to show the influence of the H2S quencher for spectroscopic response of probe and formation of NBD-SH.

Answer: Thank you very much for this suggestion. We performed additional real time NMR test to figured out the formation of the NBD-SH (Figure S7).

3) It will be good to show the probe response during the enzymatic release of H2S in real time.

Answer: Thank you very much for this suggestion. The imaging of D-Cys-induced H2S biogenesis was from the enzymatic-produced H2S in Figure 5. And the enzyme inhibitor AOAA can inhibit the D-Cys-induced H2S biogenesis, as visualized by our probe.

4) The article also lacks a comparison of properties of the described probe with other probes operating according to the same mechanism (fluorescence enhancements, reaction rate, limit of the detection).

Answer: Thank you very much for this comment. Herein, we provided a supplementary Table S1 for such purposes.

Table S1 comparison of properties of our probe with other probes operating

Probe

λex/λem (nm)

Fluorescence enhancement

ɸ

LOD/μM

Rate/K2

Ref

620/660

~10

0.29

0.27

29.8 M-1s-1

This work

730/796

~87

ND

0.04

14.9 M-1 s-1

1

565/585

~19

0.77

0.36

27.8 M-1s-1

2

567/589

~4.5

0.36

0.58

113 M-1 s-1

3

502/530

~65

0.64

0.057

28 M-1 s-1

3

449/496

~200

0.81

0.9

7.6 M-1 s-1

4

394/532

~68

ND

2.46

20.4 M-1 s-1

5

415/560

~273

ND

0.43

6.8 M-1 s-1

6

480/510

~150

ND

2.6

ND

7

330/468

~29

ND

0.024

ND

8

5) Before acceptance of this paper some technical aspects should be clarified:

Page 7; line 202 and page 8 line 216. How many milliliters of water were added? Page 7; line 204. What does PE abbreviation mean? Page 8; line 214. How much ice was added? Page 9; line 268. Maybe it's better to write "cell cultures".

Answer: Thank you very much for this comment. We have modified these texts.

Page 8; line 249. How was the Na2S concentration verified?

Answer: Thank you very much for this comment. Na2S is a well-known inorganic H2S donor that is widely employed in the study of H2S effects on physiology. It dissociates rapidly in PBS buffer (pH 7.4), leading to an instant formation of H2S: S2− + H2PO4-→ SH + HPO42-; HS + H2PO4- → H2S + HPO42-.

Page 8; line 245. What was the incubation time of analyte in solution with probe before the spectra were recorded?

Answer: Thank you very much for this comment. All reaction mixtures were shaken uniformly before spectra measurements. These actions could be completed in seconds.

Reviewer 2 Report

The manuscript prepared by Ismail et al. reports the synthesis and characterization of a rhodamine-based fluorescent probe for the detection of H2S in living cells. The authors describe a nice work however, the claims are not justified with the experimental evidences. Given the level of novelty on this study and the experimental evidences provided so far, the manuscript does not fulfill the criteria for admission on Molecules journal.

General comments:

First, I have a serious concern with claiming that the probe is a NIR emitter dye…The authors should be careful with such statements. The NIR window starts at 750-780 nm and can go up to for 1,000 nm, whereas for 2ndNIR window goes from 1,000 to 1,700 nm.

The probe proposed here is clearly not within this NIR region and, the title of the manuscript should be definitely changed to accurately reflect the findings.

Second, I find it difficult to be so affirmative about the probe interacting selectively with H2in the living cells… cells are such complex media that there could be so many other elements that could turn-ON the probe…

Third, I find it concerning to use Na2S as an equivalent of H2S, furthermore at such high concentrations (100 – 500 micromolar) as used in this study.

Major concerns:

- The authors use Na2S as an equivalent to H2S for all their measurements in vitro and in cellulo… Can the authors provide clear argument about using Na2S as an alternative to direct H2S?? What are the reactions involved with Na2S in H2O or PBS buffer? How is H2S generated? Do you generate equivalent amount of H2S from Na2S?

Sodium sulfide in water will give you sodium hydroxide…isn’t this concerning for your experiments?

- The language of using H2S throughout the text, while you use Na2S is not quite accurate.

- The thiolysis reactions was characterized by HRMS… Could you provide the full spectrum? The window of the spectrum provided in the SI is very narrow, clearly focusing only on the interested peak…

Furthermore, you can monitor this reaction also via NMR…

- The in cellulo studies are not clear, somehow contradictory too…I do not understand why do you need to use D-Cys with the probe 1 for endogenous detection of H2S?? What is different between D-Cys and Cys??

- In the text (Line 154-155), the authors state that treatment of the cells with the probe does not yield any fluorescence turn-on. Could this be due to non-cellular uptake of probe 1? When treated with 1 and Na2S they luminesce… Is the sodium sulfide treatment performed after treatment of the cells with 1?

In line 174-175 the authors claim that the probe is promising tool for imaging of H2S generated in mitochondria. From the fluorescence images, which have very weak resolution, it is difficult to conclude… It seems that the dye stains all the cell. The images do not show any specificity or selectivity for mitochondria or any other organelle. 

Minor revisions:

Line 17: its low, revise as, their low

Line 18: that involved, revise as, that is involved in

Scheme 1: the ball/stick representation of hydrogen sulfide, replace with chemical structure H2S. Also, the flesh AziRho-NBD to 1, is confusing, please remove it.

Line 80: by using procedure, revise as by using the procedure

Line 94: are illustrated in (Figure 1), revise as, are illustrated in Figure 1

Line 94: in (Figure 1A) revise as in Figure 1A

Line 95: revise as, UV absorbance maxima at… Additionally, in absorption profile there is a band at 450 nm. Where does this band come from? What transitions are involved in these bands, i.e. 450, 500 and 620 nm?

Line 98: in PBS buffer (50 mM, pH 7.4)… describe composition of the buffer, what is at 50 mM??

Figure 1 capture. Include the buffer solution used

Line 105: revise PET, as photoinduced electron transfer process (PET)

Line 106: “a large fluorescence change in NIR range was observed…” this is very bold and more importantly not accurate. Please revise. The same is true for statement in line 111.

Figure 3. In the capture include the buffer solution used, and if measured at room temperature.

Line 146: Revise as, Encouraged by the…… we moved forward to study the biological applications of 1.

Line 148: Here you indicate that the viability tests were performed after 48 h incubation with the probe, however in the SI is indicated after 24 h. Revise accordingly.

Author Response

The manuscript prepared by Ismail et al. reports the synthesis and characterization of a rhodamine-based fluorescent probe for the detection of H2S in living cells. The authors describe a nice work however, the claims are not justified with the experimental evidences. Given the level of novelty on this study and the experimental evidences provided so far, the manuscript does not fulfill the criteria for admission on Molecules journal.

General comments:

1) First, I have a serious concern with claiming that the probe is a NIR emitter dye. The authors should be careful with such statements. The NIR window starts at 750-780 nm and can go up to for 1000 nm, whereas for 2nd NIR window goes from 1000 to 1700 nm.

The probe proposed here is clearly not within this NIR region and, the title of the manuscript should be definitely changed to accurately reflect the findings.

Response 1: Thank you very much for pointing out this. Current research and clinical practice mainly utilizes near-infrared light with the wavelengths ranging from 650-950 nm (Adv. Meter., 2018, 30, 1705980). While our probe shows a wide range of fluorescence from 640-800 nm, which can be considered as an NIR probe (Anal. Chem. 2017, 89, 1922).

2) Second, I find it difficult to be so affirmative about the probe interacting selectively with H2S in the living cells. Cells are such complex media that there could be so many other elements that could turn-on the probe.

Response 2: Thank you very much for this comment. Control experiments in Figure 5 suggest that 1 could be used for selective imaging of exogenous and endogenous H2S.

3) Third, I find it concerning to use Na2S as an equivalent of H2S, furthermore at such high concentrations (100 – 500 micromolar) as used in this study.

Response 3: Thank you very much for this comment. In Figure 3, the H2S concentrations were from 2.5 – 50 micromolar.

Major concerns:

1) The authors use Na2S as an equivalent to H2S for all their measurements in vitro and in cellulo. Can the authors provide clear argument about using Na2S as an alternative to direct H2S?? What are the reactions involved with Na2S in H2O or PBS buffer? How is H2S generated? Do you generate equivalent amount of H2S from Na2S?

Response 1: Thank you very much for this comment. Na2S is a well-known inorganic H2S donor that is widely employed in the study of H2S effects on physiology. Na2S dissociates rapidly in PBS buffer (pH 7.4), leading to an instant formation of H2S: S2− + H2PO4-→ SH + HPO42-; HS + H2PO4- → H2S + HPO42-.

2) Sodium sulfide in water will give you sodium hydroxide, isn’t this concerning for your experiments?

Response 2: All tests were in PBS buffer, rather than pure water.

3) The language of using H2S throughout the text, while you use Na2S is not quite accurate.

Response 3: Thank you very much for this comment. Na2S is a well-known inorganic H2S donor in buffer: S2− + H2PO4-→ SH + HPO42-; HS + H2PO4- → H2S + HPO42-.

4) The thiolysis reactions was characterized by HRMS. Could you provide the full spectrum? The window of the spectrum provided in the SI is very narrow, clearly focusing only on the interested peak.

Response 4: According to your suggestion, we have provided the full spectrum data in the SI.

5) Furthermore, you can monitor this reaction also via NMR.

Response 5: According to your suggestion, we have performed additional experiments to monitor the change in the 1H NMR spectrum after the addition of H2S (Figure S7).

6) The in cellulo studies are not clear, somehow contradictory too. I do not understand why do you need to use D-Cys with the probe 1 for endogenous detection of H2S?? What is different between D-Cys and Cys??

Response 6: Thank you very much for this comment. D-Cys can induce H2S biosynthesis via the 3-MST pathway (Nat. Commun., 2013, 4, 1366-1372), so we can use D-Cys-treated HeLa cells with probe 1 for endogenous detection of H2S.

7) In the text (Line 154-155), the authors state that treatment of the cells with the probe does not yield any fluorescence turn-on. Could this be due to non-cellular uptake of probe 1? When treated with 1 and Na2S they luminesce. Is the sodium sulfide treatment performed after treatment of the cells with 1?

Response 7: Thank you very much for this comment. The probe is a small molecule without negatively-charged group, so it can be cell membrane-permeable. On the other hand, there is nearly no endogenous H2S in HeLa cells, so we could not observe any enhancement of fluorescence. Additionally, strong intracellular fluorescence was observed after co-incubating with D-Cys and probe 1, suggesting probe 1 entered cells and was activated by D-Cys-induced endogenous H2S.

8) In line 174-175 the authors claim that the probe is promising tool for imaging of H2S generated in mitochondria. From the fluorescence images, which have very weak resolution, it is difficult to conclude. It seems that the dye stains all the cell. The images do not show any specificity or selectivity for mitochondria or any other organelle.

Response 8: Thank you very much for this suggestion. The co-localization experiments are shown in Figure 6. The Pearson’s coefficient is 0.946 of probe 1 with Mito-Tracker Green FM, suggesting that probe 1 has good mitochondria-targeting ability.

Minor revisions:

Line 17: its low, revise as, their low. Line 18: that involved, revise as, that is involved in. Scheme 1: the ball/stick representation of hydrogen sulfide, replace with chemical structure H2 Also, the flesh AziRho-NBD to 1, is confusing, please remove it. Line 80: by using procedure, revise as by using the procedure. Line 94: are illustrated in (Figure 1), revise as, are illustrated in Figure 1. Line 94: in (Figure 1A) revise as in Figure 1A. Figure 1 capture. Include the buffer solution used. Line 105: revise PET, as photoinduced electron transfer process (PET).

9) Line 106 “a large fluorescence change in NIR range was observed…” this is very bold and more importantly not accurate. Please revise. The same is true for statement in line 111.

10) Line 146: Revise as, encouraged by the…… we moved forward to study the biological applications of 1.

11) Line 148: Here you indicate that the viability tests were performed after 48 h incubation with the probe, however in the SI is indicated after 24 h. Revise accordingly.

Response 1-11: Thank you very much for your suggestions. We have modified the texts.

12) Figure 3. In the capture include the buffer solution used, and if measured at room temperature.

Response 12: Thank you very much for this comment. All spectroscopic measurements were performed in PBS buffer at room temperature.

13) Line 98: in PBS buffer (50 mM, pH 7.4), describe composition of the buffer, what is at 50 mM??

Response 13: Thank you very much for this comment. 50 mM PBS refers to 50 mM HPO42- and H2PO4-.

14) Line 95: revise as, UV absorbance maxima at… Additionally, in absorption profile there is a band at 450 nm. Where does this band come from? What transitions are involved in these bands, i.e. 450, 500 and 620 nm?

Response 14: Thank you very much for this comment. The peaks at 450 nm and 620 nm are assigned to the extended π-conjugated rhodamine moiety (Anal. Chem. 2017, 89, 1922), while the peak at 500 nm is due to the NBD amine moiety.

Round 2

Reviewer 1 Report

Can be accepted in present form.

Author Response

We feel great thanks for your professional review work on our article.

Reviewer 2 Report

I appreciate the work of the authors in improving the quality of the manuscript. However, there are a few more points to improve and revise before acceptance for publication. Although, I still disagree with the authors about claiming the molecule as NIR probe...two papers do not counterbalance all the rest of the papers and the common knowledge about what is accepted as NIR window.

Major question: If this probe is addressed to cells that naturally express a high level of H2S, why choosing HELA cells? Why not some cells that have a mutation in one of the mentioned pathways, and express H2S in large amounts. 

Line 21: NBD amines, replace as, 7-nitro 1,2,3- benzoxadiazole (NBD) amines. This is the first time you mention NBD in the text, so you should indicate the full name along with its abbreviation.

Line 80: usingthethree-step, please revise as, using a three-step.

Line 85: The structure of cmp 1… was confirmed by….and HRMS. Please add the following sentence: “The spectra (Figure Sx, etc.) are included in the Supplemental Information.” As a general note for the authors, all the figures should be referenced in the text.

Scheme 2. The molecules seem squeezed all together. Please, separate them such as the reaction fleshes do not overlap with the chemical structures.

Line 95: In the phrase: which assignedto the rhodamine..., revise as, which are assigned…

Line 98. Please explain briefly within the text (or you may add a footnote in the reference section) about Na2S. Why is ok to use this molecule as an equivalent of H2S…

Your response is very good but not all readers know about it. Na2S is a well-known inorganic H2S donor that is widely employed in the study of H2S effects on physiology. Na2S dissociates rapidly in PBS buffer (pH 7.4), leading to an instant formation of H2S: S2−+ H2PO4-→ SH+ HPO42-; HS+ H2PO4-→ H2S + HPO42-.

Line 100: The authors included the full spectrum of HRMS characterization and included the NMR study as requested. However, the NMR study should be mentioned in the text, maybe along the HRMS study, not only incorporated in the SI. Revise accordingly.

Figure 2: The labeling of all the figures should be consistent. In Figure 2, instead of a), b) use A, B, etc.. along the figures and the figure capture.

Figure 3. Label the figures as A, B, C and update the figure capture and the text (Line 117 – 120) accordingly.

Line 148: Encouraging by, revise as, Encouraged by…

Line 158: “For detection of endogenous production of H2S, cells were co-incubated with D-Cys and 1.” Please, incorporate the comments (one sentence or two) about the role of CysD in the text, otherwise it is difficult for the readers to understand the utility of CysD on these experiments. And a reference for that…Response 6 included in the text. D-Cys can induce H2S biosynthesis via the 3-MST pathway (Nat. Commun.,2013, 4, 1366-1372), so we can use D-Cys-treated HeLa cells with probe 1for endogenous detection of H2S.

In the supplemental Information, include a table of content or figures. All the figures should be numbered including the NMR and MS spectra. The references should be indicated at the end of SI. The pages of the supplemental info are usually numbered as S1, S2, S3 etc. instead of 1, 2, 3…

Author Response

Response to Reviewer 2 Comments

I appreciate the work of the authors in improving the quality of the manuscript. However, there are a few more points to improve and revise before acceptance for publication. Although, I still disagree with the authors about claiming the molecule as NIR probe...two papers do not counterbalance all the rest of the papers and the common knowledge about what is accepted as NIR window.

Major question:

1) If this probe is addressed to cells that naturally express a high level of H2S, why choosing HeLa cells? Why not some cells that have a mutation in one of the mentioned pathways, and express H2S in large amounts.

Response 1: Thank you very much for this comment. D-Cys can induce H2S biosynthesis via the 3-MST pathway (Nat. Commun., 2013, 4, 1366-1372). And the comparison of imaging of HeLa cells with or without treatment with D-Cys can verify the specific imaging of the probe with respect to H2S (Figure 5).

2) Line 21: NBD amines, replace as, 7-nitro 1,2,3- benzoxadiazole (NBD) amines. This is the first time you mention NBD in the text, so you should indicate the full name along with its abbreviation.

Line 80: usingthethree-step, please revise as, using a three-step.

Line 85: The structure of cmp 1… was confirmed by….and HRMS. Please add the following sentence: “The spectra (Figure Sx, etc.) are included in the Supplemental Information.” As a general note for the authors, all the figures should be referenced in the text.

Scheme 2. The molecules seem squeezed all together. Please, separate them such as the reaction fleshes do not overlap with the chemical structures.

Line 95: In the phrase: which assigned to the rhodamine..., revise as, which are assigned…

Response 2: According to your suggestions, we have modified these in the main text and SI.

3) Line 98. Please explain briefly within the text (or you may add a footnote in the reference section) about Na2S. Why is ok to use this molecule as an equivalent of H2S…

Your response is very good but not all readers know about it. Na2S is a well-known inorganic H2S donor that is widely employed in the study of H2S effects on physiology. Na2S dissociates rapidly in PBS buffer (pH 7.4), leading to an instant formation of H2S: S2−+ H2PO4-→ SH+ HPO42-; HS+ H2PO4-→ H2S + HPO42-.

Response 3: According to your suggestion, we have added a description in Line 98.

4) Line 100: The authors included the full spectrum of HRMS characterization and included the NMR study as requested. However, the NMR study should be mentioned in the text, maybe along the HRMS study, not only incorporated in the SI. Revise accordingly.

Response 4: According to your suggestion, we have revised in Line 101.

5) Figure 2: The labeling of all the figures should be consistent. In Figure 2, instead of a), b) use A, B, etc.. along the figures and the figure capture.

Figure 3. Label the figures as A, B, C and update the figure capture and the text (Line 117 – 120) accordingly.

Line 148: Encouraging by, revise as, Encouraged by…

Response 5: According to your suggestions, we have modified these in the main text.

6) Line 158: “For detection of endogenous production of H2S, cells were co-incubated with D-Cys and 1.” Please, incorporate the comments (one sentence or two) about the role of D-Cys in the text, otherwise it is difficult for the readers to understand the utility of D-Cys on these experiments. And a reference for that…Response 6 included in the text. D-Cys can induce H2S biosynthesis via the 3-MST pathway (Nat. Commun.,2013, 4, 1366-1372), so we can use D-Cys-treated HeLa cells with probe 1 for endogenous detection of H2S.

Response 6: According to your suggestion, we have added a description in Line 161.

7) In the supplemental Information, include a table of content or figures. All the figures should be numbered including the NMR and MS spectra. The references should be indicated at the end of SI. The pages of the supplemental info are usually numbered as S1, S2, S3 etc. instead of 1, 2, 3…

Response 7: According to your suggestion, we have modified all the content in SI.
